# Assessment of Forest Road Models in Concession Areas in the Brazilian Amazon

**Pricila do Prado Morais [1,\*], Eugenio Yatsuda Arima [2], Álvaro Nogueira de Souza [1], Reginaldo Sérgio Pereira [1], Fabiano Emmert [3], Rodrigo Montezano Cardoso [1], Eder Pereira Miguel [1] and Eraldo Aparecido Trondoli Matricardi [1,\*]**

[1] Graduate Program in Forestry Sciences, Department of Forestry Engineering, University of Brasília/UnB, University Campus Darcy Ribeiro, Asa Norte, Brasília, DF 70910-900, Brazil; ansouza@unb.br (Á.N.d.S.); reginaldosp@unb.br (R.S.P.); montezano_florestal@yahoo.com.br (R.M.C.); miguelederpereira@gmail.com (E.P.M.)

[2] Department of Geography and the Environment, University of Texas at Austin, 305 E. 23rd Street, Austin, TX 78712, USA; arima@austin.utexas.edu

[3] Institute of Agrarian Sciences/ICA, Federal Rural University of Pará/UFRA, Estrada Principal, Curió-Utinga, Belém, PA 66610-770, Brazil; fabianoemmert@gmail.com

\* Correspondence: pricila.pradomorais@gmail.com (P.d.P.M.); ematricardi@unb.br (E.A.T.M.)

**Abstract:** Forest management aimed at the sustainable use of forest resources is an alternative land use to deforestation and can improve forest conservation in tropical regions. The construction of forest infrastructure, including forest roads, skid trails, and log-landings, is a key factor in minimizing the impacts and forest disturbances typically caused by selective logging activities in tropical forests. In this study, we used field and secondary data to assess the planned and implemented forest infrastructure in a study site of 5723 hectares under a forest concession in the Caxiuanã National Forest, located in the state of Pará, Brazilian Amazon. We tested alternative modeling approaches (the Tomlin and Spanning Tree models) by comparing them with the previously planned and implemented logging infrastructure by a concessionaire timber company (CEMAL Ltd.) in the study site. Our results indicate that the Tomlin model was the best approach for allocating forest roads in the study area, as it demonstrated the optimal balance between financial costs and forest disturbances for timber extraction. Additionally, Minimum Spanning Tree modeling achieved the most favorable results in delineating secondary roads and skid trails in the study site, despite slightly higher financial costs compared to the minimum acceptable costs. This alternative approach to modeling forest infrastructure can contribute to reducing forest disturbances and increasing the economic and ecological sustainability of forest management in tropical forests.

**Keywords:** Amazon; sustainable forest management; forest concession; forest roads

## 1. Introduction

Timber production, aimed at supplying national and international markets, is one of the main economic activities in the Brazilian Amazon. The region is home to several tree species with high commercial value and suitability for wood production, making it one of the largest tropical timber-producing regions in the world [1]. In 2018, Brazil reported a tropical timber production of 10,745,178 m$^3$, with the majority coming from the state of Pará, the largest exporter of tropical timber in Brazil [2].

Forest management for timber production is an alternative for the sustainable use of forest resources in tropical forests [3–7]. In the Brazilian Amazon, forest concessions (a type of forest leasing in Brazil) established by the Brazilian Federal Law no. 11,284 in 2006, allow concessionaires to explore forest products from National Forests, in accordance with previously approved Sustainable Forest Management Plans (SFMP) [8].

Forest management requires the implementation of forest infrastructure for accessing and conducting logging activities, including primary forest roads (main forest access

routes), secondary forest roads (connecting log-landings to main forest roads), and skid trails. This type of infrastructure can cause unnecessary damage to the forest without proper planning [9–15].

Various techniques have been adopted by timber companies for selective logging in tropical forests, with the aim of reducing impacts on the remaining forest [7,16,17]. The planning phase and initial design of logging infrastructure are crucial in reducing financial costs, time, and the environmental impact caused by selective logging activities in tropical forests [18]. The most suitable forest road network must consider distances between commercial trees to be harvested and log-landings, considering terrain conditions and forest structure [19].

However, planning road networks for timber extraction is a complex and time-consuming task [20], and it remains a great challenge to properly plan forest infrastructure aiming to minimize forest impacts [15]. Mathematical techniques and computer algorithms that automate the design of a forest road network, such as linear programming and network design algorithms based on graph theory, can improve the range of tools available to support improved forest planning [2,21–23].

In this research, we aimed to develop and test alternative approaches to designing forest roads and skid trails using the Tomlin and Minimum Spanning Tree approaches. To accomplish this, we analyzed planned and field-implemented forest infrastructure in a forest management project conducted by CEMAL Ltd., a timber company currently operating in the Caxiuanã National Forest, Pará state, Brazil. Alternative designs of road and skid trail networks were developed using Tomlin's [21] and Minimum Spanning Tree algorithms [24,25]. We then compared these different model approaches for building forest infrastructure in the study area with the planned and actual networks constructed by the company, using economic and environmental criteria. Our study results can provide support for forest road network planning, contributing to the reduction in impacts caused by logging activities and increasing the economic viability of timber production in tropical regions.

## 2. Materials and Methods

### 2.1. Study Site

The study site encompassed a total of 5729.5 hectares within a forest concession area of 317,946 hectares in the Caxiuanã National Forest, created by the Federal Decree no. 239 of 28 November 1961, in the municipalities of Portel and Melgaço, state of Pará, Brazil (Figure 1). In the case of Caxiuanã National Forest, 59% of its territory is spatially located within the municipality of Portel and 41% within the municipality of Melgaço [26].

All National Forests in Brazil are classified as a type of protected area oriented towards various forms of sustainable forest use, which can be granted through forest concessions provided by the Brazilian Forest Service. These concessions encompass selective logging activities that can be implemented following prior analysis and approval by the National Forest management board, in accordance with the legal instruments that establish them [26].

Complementarily, our study site encompassed three Annual Production Units (APUs) within the forest concession area that were selectively logged between 2019 and 2021 (Table 1). Each APU is defined as a forest area capable of supplying the forest concessionaire for an entire year. The field datasets and reports related to selective logging activities were properly stored and provided by CEMAL Ltd., the timber company granted responsibility for enforcing selective logging in the Caxiuanã National Forest through a previously approved forest management plan by the Brazilian National Forest Service [27].

According to the Köppen climatic classification, the climate of the Caxiuanã National Forest is hot and tropical humid, characterized by a short dry season. The average air temperature is 26.7 °C, with a minimum temperature of 22 °C and maximum of 32 °C. Although the Amazon rainforest region does not experience a pronounced dry season, it does have a well-defined annual dry period. The rainy season typically occurs between December and June, while the dry season spans from August to November. July commonly

serves as a transition month between the end of the wet season and the beginning of the dry season [28]. The study site shows a homogenous topographic form, predominantly flat in a low-relief plateau, altitude varying from 40 to 80 m [27].

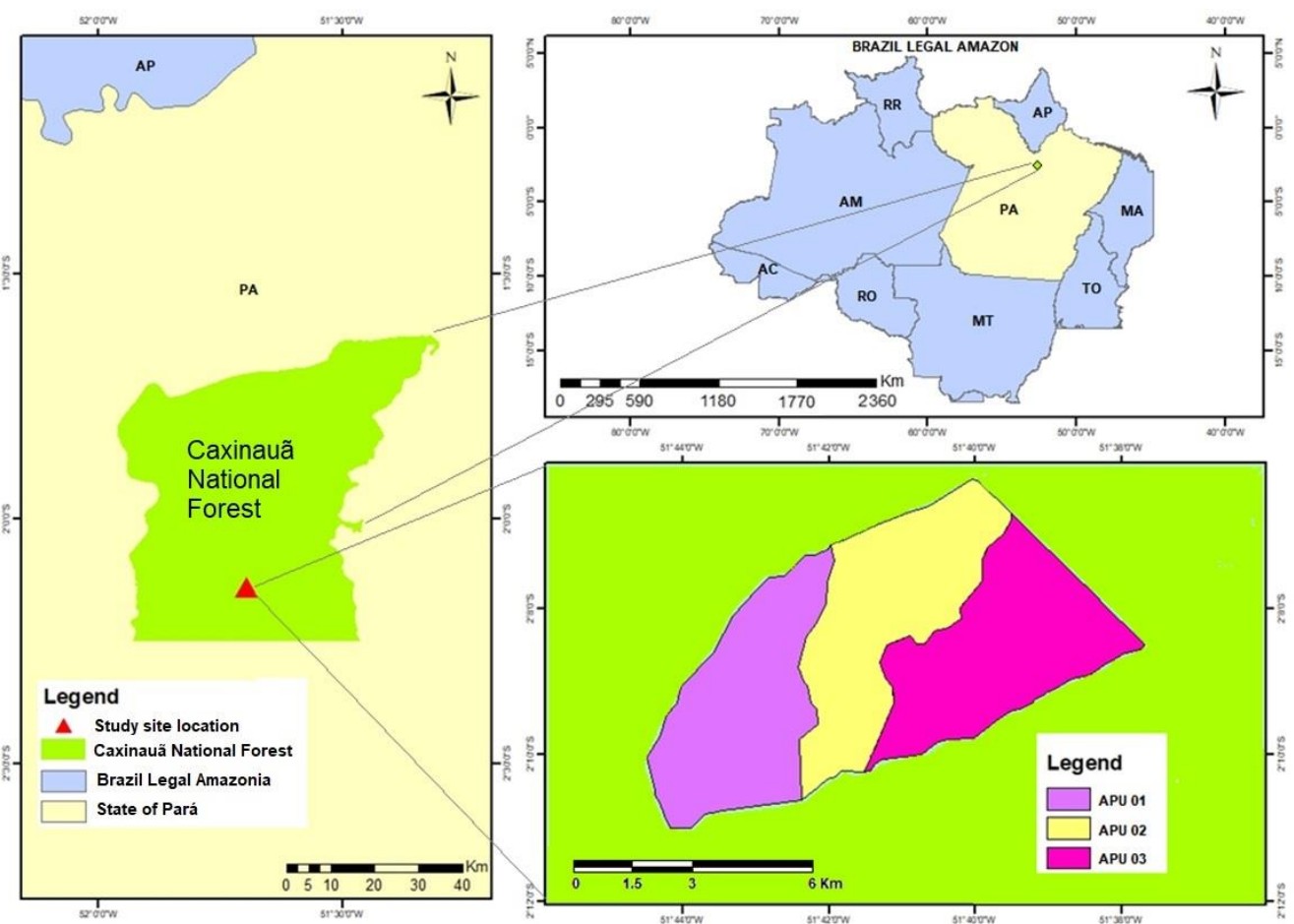

**Figure 1.** The study site is situated within the Caxiuanã National Forest, located in the state of Pará, Brazil. It includes three Annual Production Units (APUs) within the forest concession area, which were granted to the CEMAL timber company by the Brazilian Forest Service.

**Table 1.** Annual Production Units (APU) in the study site within the Caxiuanã National Forest, state of Pará, Brazil.

| Annual Production Unit (*) | Area (ha) | Harvesting Year |
|---|---|---|
| 1 | 1828.5 | 2019 |
| 2 | 1951.9 | 2020 |
| 3 | 1949.1 | 2021 |
| **Total** | **5729.5** | |

* APU is defined as a forest area capable of supplying the logging concessionaire for an entire year.

Based on the vegetation map provided by the Brazilian Institute of Geography and Statistics (IBGE) for the Legal Amazon, the Caxiuanã National Forest is predominantly covered by lowland ombrophilous dense forest. According to the forest inventory dataset provided by ICMBIO (2012), a total of 138 tree species have been identified, exhibiting an average basal area of 28.1 $m^2.ha^{-1}$ and a volume of 249.1 $m^3.ha^{-1}$ for individual trees showing Diameter at Base Height (DBH) greater than 10 cm. The Caxiuanã National Forest exhibits high local diversity, with a Shannon–Wiener (H) index of 4.5. The potential timber

production was estimated to be approximately 26 individual trees and 112 $m^3.ha^{-1}$ for 96 commercial tree species with a DBH greater than 50 cm [26].

*2.2. Field Datasets*

A forest concession agreement was granted to CEMAL Ltd., a timber company, in 2016 by the Brazilian Forest Service, and it will last for 40 years. This agreement allows the company to implement forest management practices in the Caxiuanã National Forest, located in the state of Pará, focusing on reduced impact selective logging of native tree species. For this analysis, we utilized datasets provided by [26] for three Annual Production Units (APUs) that were selectively logged between 2019 and 2021.

The CEMAL timber company conducted a census forest inventory within each of the Annual Production Units (APUs), following the official standards and methodological approaches issued by the Brazilian Institute of Environment and Renewable Natural Resources (IBAMA). This inventory resulted in field datasets containing dendrometric measurements, tree classifications and locations, and site variables, providing valuable information to support the forest planning process for selective logging, reported by [27].

Based on the field inventory, the timber company operating under the concession has identified ten primary commercial tree species that showed higher commercial value. These species include Muiracatiara (*Astroninum lecointei*), Ipê (*Tabebuia* spp.), Angelim-Vermelho (*Dinizia excelsa*), Cumaru (*Dipteryx* spp.), Jatobá (*Humenaea courbaril*), Angelim (*Hymenolobium* spp.), Cupiuba (*Goupia glabra*), Itauba (*Mezilaurus* spp.), Tatajuba (*Bagassa guianensis*), and Maçaranduba (*Manilkara* spp.) [26].

We used the original plan of forest infrastructure (log-landings and forest roads) and the forest infrastructure implemented in the study site by the CEMAL timber company. Spatial locations of trees, patios, and forest roads were collected in the field using navigation GPS devices and the export-to-vector format for processing in a GIS computer environment. The spatial location of the contour lines at 10 m intervals, APU boundaries, commercial logged trees, primary and secondary forest roads, and Permanent Protection Areas were also collected in the field and used in this analysis.

The logging companies are responsible for conducting forest management activities in an appropriate manner, adhering to a previously approved forest plan that specifies the location and size of log-landings. These landings serve as areas where logs, dragged by skidders from the forest, are grouped and measured [29]. The log-landings are interconnected by secondary and main roads, facilitating the transportation of the logs to the sawmills using heavy-loaded timber trucks [30].

Figure 2 shows the spatial location of the harvested trees, the Permanent Protected Areas (PPA), and the log-landings within each Annual Production Unit (APU) in the study site.

Originally, the criteria used for the allocation of forest roads and log-landings by the timber company responsible for implementing the forest management plan in the study area were based the field data collected during the forest inventory and by defining the buffer distance of each log-storage patio (defined by a circle of 25 m radius). Those patios were evenly distributed throughout the logging area, showing a high potential of commercial trees to be harvested. Subsequently, the Digital Elevation Model (DEM) provided by Shuttle Radar Topography Mission (SRTM) was used to estimate the slope throughout the Annual Production Units. That information was combined to support the definition of the main road locations, which were mostly located in flat areas or lower slopes or in higher elevations, aiming to connect the log-landings in the study area. In addition, secondary roads were included to facilitate access to log-landings, prioritizing the shortest distance and avoiding impacts on the PPA. Figure 3 illustrates the forest roads, log-landings, and tree trunks just after logging in the Caxiuanã National Forest.

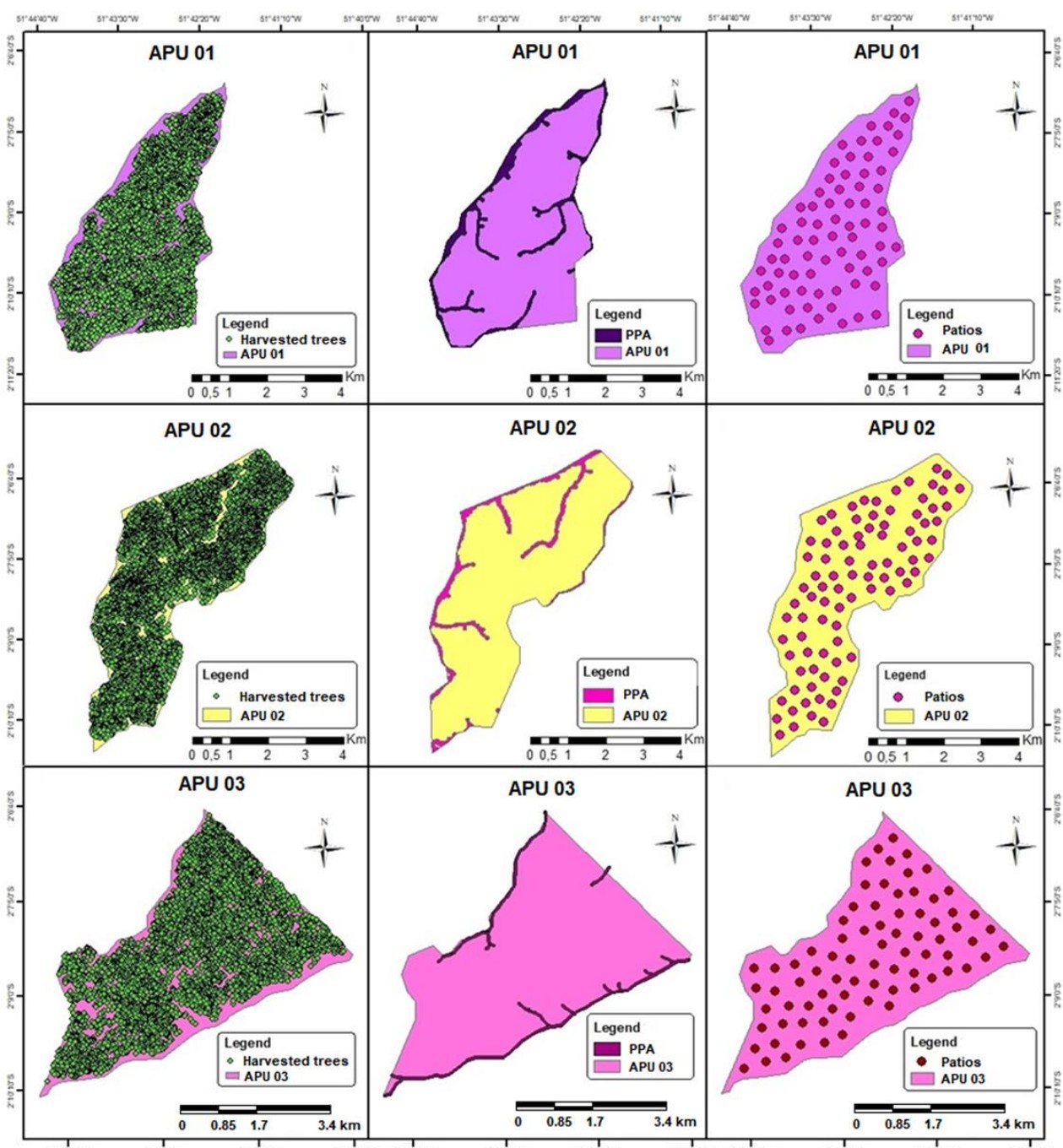

**Figure 2.** Spatial location of the logged trees, the Permanent Protected Areas (PPA), and the log-landings within each Annual Production Unit (APU) in the study site.

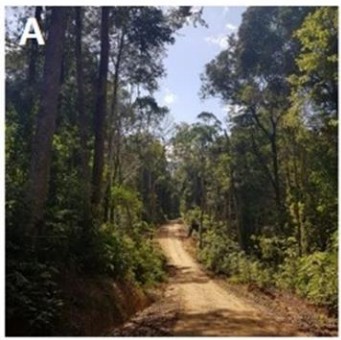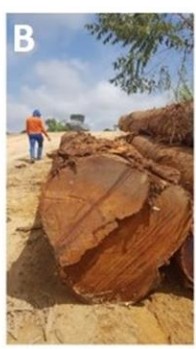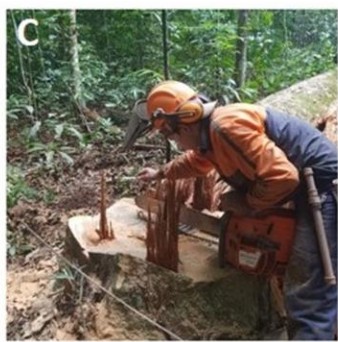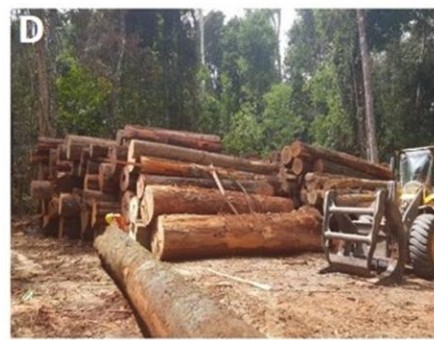

**Figure 3.** (**A**) Forest road; (**B**) trees dragged from the forest; (**C**) a tree trunk just after felling; and (**D**) a log-storage patio in the Caxiuanã National Forest, state of Pará, Brazil. Photos were taken in 2019 and provided by [27].

*2.3. Forest Road Modeling*

For modeling the forest roads in the study area, we used maps containing the Permanent Protection Areas, the slope derived from the DEM (Digital Elevation Model), the location of log-landings, and the location of the harvested trees in the study site. The applied model approach was developed by [31], aiming to define the optimal location (road layout) by calculating the least cost path and showing more economic and environmental benefits for planning selective logging activities.

The optimization of the spatial road network is a problem known in the mathematics and computing fields as computational graph theory [24]. The logging fields would correspond to the vertices in graph theory, and the segments of the roads to the edges. The goal of optimization is, therefore, to connect the log-landings through a road network formed by segments in such a way that the total costs, according to a defined metric, are minimized. In this regard, several algorithms have been developed by mathematicians and computer scientists over the centuries to address this problem. This seemingly simple problem is one of the most difficult problems in mathematics, and there is no algorithm that guarantees an optimal solution in a finite time when the number of vertices is large.

For simplicity, consider a uniform plane with one origin at the bottom and two destinations, as depicted in Figure 4A. A simple Least Cost Path (LCP) analysis using GIS built-in functions would return a solution where each point is individually connected to the origin by its LCP, which in this case is a straight line, given that the plane is uniform. However, this is not the global solution, which is known as the Steiner Problem, and which is illustrated in Figure 4B. Notice that by including another connecting point in the middle, the overall network can be reduced in size. This problem, however, is not easy to solve when the number of points is large. In computer science, the Steiner Problem is an NP-problem (non-deterministic polynomial problem): the most difficult class of problems. Therefore, some heuristic algorithms have been developed which do not guarantee the global minimum solution but are better than the solution obtained with the Least Cost Path of ArcGIS® (Figure 4A).

In this analysis, we employed the Least Cost Path (LCP) technique, which incorporates the Dijkstra algorithm [32] as a subroutine to calculate individual segments. The calculation of segments connecting all logging patios to the origin was not performed in a single iteration. Instead, we employed a step-by-step approach with multiple iterations, where the source was modified by incorporating the added segment from the previous iteration.

Complementarily, two heuristic algorithms were used in this analysis. The first is called "Tomlin" (which is the name of the author who first proposed the application of this algorithm in Geographic Information Systems) [21], and the second is a modification of the "Minimum Spanning Tree" or minimum expansion tree. The Tomlin modeling approach incorporates hydrology functions to calculate segments.

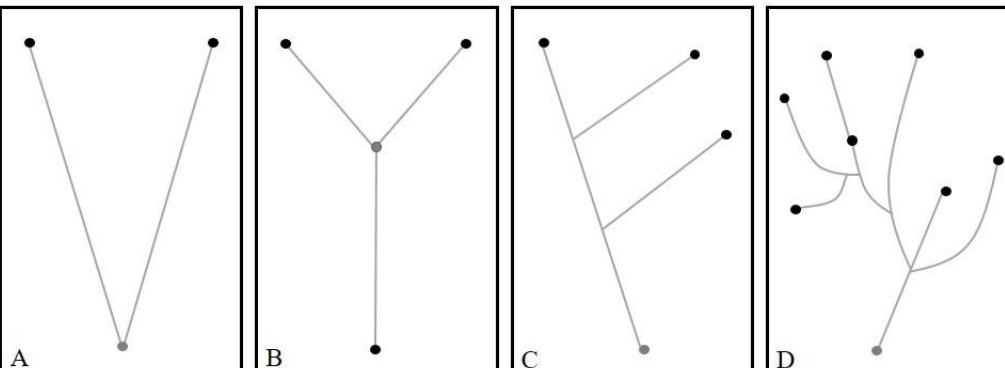

**Figure 4.** Different ways to connect dots: (**A**) solution found (Least Cost Path) in most of Geographic Information Systems; (**B**) Global Model; (**C**) Minimum Spanning Tree approach; and (**D**) Tomlin approach.

In the Tomlin model, each segment in the Least Cost Path (LCP) analysis can be compared to a river reach in hydrology, where segments are connected to one another as they flow downstream. To represent this connectivity, the Shreve classification system [33] was used. In this system, the uppermost segments are assigned a value of one, and as segments join with others downstream their values are added in succession. These values in the context of the current application represent the "hauling traffic", which indicates the number of times each segment is used to transport trees from a patio to the origin.

Next, the segment with the highest traffic is added to the previous 'origin' for the subsequent iteration; in Figure 5, the segment that is a route for 8 patios becomes a new origin. In each iteration, the algorithm selects the segments with highest 'hauling traffic', which is added to the network until all patios are connected to the network. Tomlin's algorithm works well when the friction costs are varied, which forces the segments to converge, just like water converges to form streams in rugged terrain.

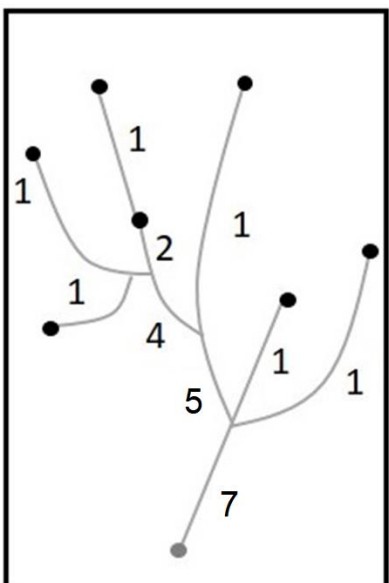

**Figure 5.** Tomlin Modeling (adapted from [21]) shows the frequency of log transportation on each forest road segment.

The second applied modeling approach was a modification of the Minimum Spanning Tree (MST) that connects road segments in stages. First, the initial origin (source) is connected to the 'closest' patio in the cost metric (i.e., weighted distance). This segment becomes a new source from which the accumulated cost will be calculated, and the next segment with the least accumulated cost will be connected to the network. This process

is repeated until all logging patios are reached. In its original inception, the MST allowed connection only at the vertices (i.e., patios in our application). We modified the algorithm to allow all points along the line segment to serve as potential connecting points (Figure 4C).

Two friction costs were used in our modeling effort. First, we used Tobler's hiking formula, originally developed to estimate the number of hours a hiker would spend moving over a terrain. This friction cost was entered as a vertical factor correction in the software ArcPro and tend to produce straight least cost path lines if the terrain is not rugged from cell to cell. To capture the impact of small changes in elevation, we also ran a model where we assigned friction costs, that we called Dheight (short for delta or difference in height), which are proportional to changes in elevation between cells [30]. Therefore, the minimization procedure used in least cost path analysis will create paths that tend to follow contour lines because, by definition, elevation along contour lines does not change, yielding costs close to zero. The Least Cost Paths (LCPs) thereby produced are longer in terms of distance and not as straight and directly connected to destinations when compared to Tobler's.

Data Preprocessing:

The least cost paths calculated in each iteration require several raster and vectors as inputs. In the first iteration, we defined the origin as a point at the border of the study area that intersects with an already existing road that is used to bring the logs from the APU to a processing mill. The 'destination' was a point vector file containing all the patios that were to be connected.

To avoid road segments in permanent protected areas (APP), we masked out the cells that contained APPs from the DEM. This forced the logging roads to completely avoid these areas. Both Tomlin's and MST's algorithms were implemented in a Python script using ArcPro's libraries because the software handles georeferenced data and contains built in hydrology (e.g., flow direction and accumulation, Shreve stream classification) and least cost path functions (e.g., path analysis) used in the subroutines.

### 2.4. Modeling Assessment

The planned and executed and two alternative models were compared for the study area, including the main and secondary forest roads in linear meters. Potential environmental disturbances related to the bridge construction and PPA affected by the road construction were also assessed. Finally, the optimum and acceptable densities of the road network were considered in addition to the costs of road construction, cost of tree harvesting, degradation costs, and total costs. The total area impacted by the road construction of each model was calculated and, additionally, we considered how many bridges were built and the total area of PPA impacted by the road construction.

### 2.5. Assessment of Selective Logging Operations

2.5.1. Average Skid Distance (ASD)

Ref. [34] described how a region cut by parallel roads will have a maximum log skidding distance equal to half of the distance between roads, where the shortest drag distance equals zero, so the Average Skid Distance (ASD) is a quarter of the distance between the two parallel roads. In this study, the ASD was calculated to compare the road density of each model evaluated, and was associated with the Optimal Road Separation, which is the path that the skidder will travel from the tree location to the log-storage patio. The authors estimated the relationship between road density and the average harvesting distance according to Equation (1):

$$\text{ASD} = (2500 \times \text{V} \times \text{T})/\text{RD} \tag{1}$$

where ASD = Average Skid Distance (m); RD = Road Density (m/ha); T = Correction factor for harvesting in cases where the log skidding is not a straight line nor a perpendicular line to the forest road and the ending point is not the closest to the origin of travelling point; and V = Correction factor to the road network, applied when the roads are curve and not parallel each other, showing variable distances among them. In this analysis, we applied

the VT correction factor of 1.85 because of the relationship between V (1.25) and T (1.48) (Equation (2)):

$$VT = T \times V \tag{2}$$

2.5.2. Operational Costs

The cost analysis was based on the methodological approach proposed by FAO/ECE/KFW and applied by [35], based on the accounting method, using estimated and observed values provided by [27].

The costs were classified as fixed costs (depreciation, interest, insurance, and operating personnel cost), variable costs (fuel, lubricants and greases, undercarriage, and maintenance), and administrative costs. We also used the average of the official currency conversion between BRL (Brazilian real) and USD (United States dollar) currency rates applied to September 2022 by the Brazilian Central Bank (BACEN).

Fixed Costs:

The Fixed Costs did not vary regardless of the amount of operating hours (SOUZA, 2016) and included:

(a)  Depreciation:

The straight-line depreciation was calculated by applying Equation (3), in which:

$$DP = ((Va - Vr)/(Vu \times He)) \tag{3}$$

where DP = straight-line depreciation of machinery (USD/ha); Va = The acquisition price of the machines + implements (USD); Vr = The residual values or resale value of the machines + implements (USD); Vu = The estimated lifespan (in years); and He = annual effective usage hours (hours), estimated according to the Equation (4). In this study, we assumed that Vr = 20% of the Va.

$$He = (Nd \times d \times Nt \times (100 - DT))/100 \tag{4}$$

where He = effective working hours per year (hours); Nd = number of work days per year; d = working shift duration (hours); Nt = number of work shifts per day; and DT = delays and unproductive days (%).

(b) Interests

Interest rates were calculated by applying an interest rate to the Average Annual Investment (AAI), which is equivalent to the opportunity cost that would be applied to the capital (29), calculated according to the following equation.

$$AAI = ((Va - Vr) \times (N + 1))/2 \times Vu) + Vr \tag{5}$$

where Va = machinery acquisition price + implements (USD); Vr = residual or resale value of the machine + implement (USD); and Vu = estimated lifespan (in years).

Based on this, the interest rates were generated by considering an annual interest rate of 12% using Equation (6).

$$I = Cc/2 \times i/100 \tag{6}$$

where I = annual interest per linear meter of road (USD/m); Cc = construction cost per linear meter of road (USD/m); i = annual interest rate (%).

(c) Insurances

Insurance costs are the expenses incurred by owners due to the possession or use of their machinery and the potential losses they may face during working time. The total expenditure for insurance was provided by the CEMAL timber company. The Total Fixed Costs (TFC) were calculated by adding costs of depreciation, interest, and insurances.

Variable costs:

(a)    Fuel

According to Santos (2013), fuel costs are determined based on the fuel consumption, which is calculated using Equation (7):

$$FC = Pu \times C \tag{7}$$

where FC = fuel cost (USD/he); Pu = price per liter of fuel—diesel (USD/L); and C = fuel consumption per effective working hour (L/he).

(b) Hydraulic oil

The cost of hydraulic oil is calculated in the same way as the cost of fuel, which is determined by multiplying the price by the consumption.

(c) Motor oil

The calculation for motor oil is performed in the same manner as the fuel cost calculation: price × consumption.

(d) Maintenance and Repairs

The maintenance and repair costs are calculated by dividing the acquisition values (USD) of the skidder over its lifespan in years.

(e) Tires

The cost of tires is calculated by multiplying the acquisition price of a tire by the number of tires divided by the tire lifespan.

(f) Skidder Operator (SO)

The cost of a skidder operator is calculated by adding the salary of skidder operators to the expenses associated with social charges, which include fees and taxes.

(g) Transportation Cost (TC)

The transportation cost is calculated as 3% of the total value of the variable costs.

(h) Administration Cost (AC):

The administration cost includes the expenses related to office work and supervision of field operations. In this analysis, we have considered it to be 5% of the total costs.

(i)    Total operational costs (TOC):

The total operational cost (TOC) was calculated by summing the permanent, variable, and administrative costs. The production cost was obtained by dividing the TOC by the machinery productivity.

2.5.3. Optimum Road Density (ORD)

The optimum road density was determined by utilizing the equations of global costs, specifically referring to Equation (8).

$$GC = Croad + Charv + Cdeg \tag{8}$$

where GC = Global Cost; Croad = cost of forest roads (USD/m); Charv = cost of forest harvesting (USD/m$^3$); and Cdeg = degradation cost (USD/m$^3$).

The optimal road densities were defined by employing specific mathematical formulas and were visualized in a graph. The graph consisted of four curves: (1) the Log-extraction Cost curve; (2) the Road Cost Curve; (3) the Degradation Cost curve; and (4) the Total Cost Curve. The road densities (RD) varied from 1 to 100 to analyze their impact.

Cost of forest roads

This was calculated according to the road densities by using the average annual increase in roads [18]. The cost of construction and maintenance of secondary roads provided by the CEMAL timber company was USD 1738.00, estimated using Equation (9).

$$\text{Croad} : \text{CAr}/\text{AAI} \times \text{RD} \tag{9}$$

where Croad = cost of roads (USD/m); CAr = annual cost of forest roads (USD/m); AAI = average annual increase ($m^3$/ha); and RD = road density (m/ha).

Cost of selective logging

The cost of selective logging was calculated based on the operational cost of each piece of equipment and operational productivity, applying Equation (10):

$$\text{C ext} = \text{CO}/\text{Pr} \tag{10}$$

where Cext = forest extraction cost (USD/$m^3$); CO = machinery operating cost (USD/h); and Pr = average effective productivity of the operation ($m^3$/h).

Cost of degradation

Calculated according to the following Equation:

$$\text{Cdeg} = \text{VT} \times \text{RD} \tag{11}$$

where Cdeg = degradation cost (USD/$m^3$); VT = correction factor of 1.85; and RD = road density (m/ha).

Optimum road density

The minimization of global costs can be mathematically met by the derivative of the overall cost, depending on the road density [18], defined by Equation (12).

$$\text{CG} = \text{Croad} + \text{Charv} + \text{Cdeg} \tag{12}$$

where CG = overall cost; Croad = cost of forest roads (USD/m); Charv = forest extraction cost (USD/$m^3$); and Cdeg = degradation cost (USD/$m^3$).

### 2.5.4. Acceptable Road Density

For the calculation of the Acceptable Road Density (ARD), we used the methodology developed by [36]. We applied a percentage of 5% above the minimum total cost, where the costs of reconstruction and maintenance, log extraction (harvesting), and loss of productive area also varied according to the total cost.

### 2.5.5. Optimum Separation of Secondary Roads

Optimum Separation among Secondary Roads (OSR) was calculated because it is the theoretical optimal distance that will allow the achievement of the ideal log skidding distances according to each piece of harvesting equipment (in our case, a skidder) meeting the condition of lower costs of use of equipment with lower costs of road construction [19]. The calculation of the OSR can be estimated as follows:

(a) A total of 5 to 10 average skid distances (ASD) are assumed in which the skidder will potentially work (under the relief conditions), e.g., 100, 150, 200, 300, to 1000 m, and the log skidding cost is calculated on those road lengths.

(b) By using a simplified formula 10,000/4*ASD, the corresponding optimum road densities (ORD, m/ha) are estimated for the log skidding distance. The ASD will increase as the ORD decreases.

(c) The harvesting costs in relation to the log skidding distance (considering the type of equipment) and the corresponding cost of road construction, according to the road extent per hectare (considering the standard or category of the roads), were estimated. The harvesting costs increase as the log skidding distances increase, and the road

construction costs decrease (for example, roads can be separated into two categories: permanent and temporary roads).

(d)   The corresponding costs (ASD + ORD) are added.

(e)   There will be some intermediate point between 100 m and 1000 m, where the sum will be the smallest one, which will be the optimum distance. Out of that distance there will be constructed too many or too few roads and log skidding at larger or shorter distances, normally resulting in higher costs of roads or harvesting. Because the theoretical ORD is equal to OSR/4, the distance among the selected roads will be equal to the selected ORD multiplied by 4.

### 2.5.6. Assessment of Skid Operations

We estimated the mechanized log skidding operations using a skidder forestry tractor, with productivity ($m^3 \cdot h^{-1}$) and operating (USD $h^{-1}$) and production (USD $m^{-3}$). We used a dataset (Table 2) provided by [27] including technical information and costs. We also established basic assumptions based on scientific and specialized literature to determine some of the cost components.

**Table 2.** Skidding log dataset.

| Skidding Log | | Skidder Costs | |
|---|---|---|---|
| Extracted logs per day * | 5.00 | Acquisition price (USD) * | 386,220.00 |
| Average log volume ($m^3$) * | 2.5 | Resale value (USD) ** | 77,243.20 |
| Working hours per day * | 8.00 | Estimate lifespan (year) ** | 25.00 |
| | | Fuel consumption ($L \cdot h^{-1}$) * | 14.00 |
| | | Diesel price (USD/L) | 1.35 |
| | | Hydraulic oil consumption ($L \cdot h^{-1}$) ** | 0.21 |
| | | Hydraulic oil price (USD/L) | 6.76 |
| * Dataset provided by [27] | | Motor oil consumption ($L \cdot h^{-1}$) ** | 0.13 |
| ** Data from current scientific literature | | Motor oil price (USD/L) | 3.28 |
| | | Tire price (USD/unit) ** | 2793.10 |
| | | Tire lifespan (hour)** | 2500.00 |
| | | Wage (USD/months) ** | 391.60 |

The CEMAL timber company provided all datasets related to the acquisition price of the skidder used in the logging operations (USD 386,220.00) and the diesel price (1.35 USD/L). The other values in Table 2 were calculated according to the previously presented and discussed Equations.

## 3. Results

### 3.1. Road Modeling Based on Location of the Log-Storage

The total area effectively (field-implemented) impacted by the road construction in the study site (three Annual Production Units—APUs) was slightly higher than the roads originally planned by the timber company (Table 3). The theoretical modeling also showed the total area impacted by the modelled roads as higher than originally planned by the concessionaire timber company in all three study APUs, using the field-implemented log-landings as theoretical (reference) destinations.

More specifically, we observed that the planned main roads impacted a total of 7.32 ha, while the effective (executed) main roads impacted a total area of 8.14 ha in the APU 01. Similarly, the Tomlin modeling using Tobler´s hiking formula impacted a total area of 12.09 ha, and the Minimum Spanning Tree modeling impacted a total area of 14.14 ha (Table 3, Figure 6). Based on this, we observed that the Tomlin modeling approach showed better results than the Minimum Spanning Tree modeling approach that uses the Dheight friction costs. By comparing the total area impacted by all analyzed main road construction approaches, the road network implemented by the concessionaire timber company in the Caxiuanã National Forest showed better results, with lesser extent of impacts on native forests.

**Table 3.** Total area of main roads and secondary roads in each Annual Production Unit (APU), the number of bridges, and the impacted areas on Permanent Protected Areas (PPA).

| APU * | Models | Area (Hectares) | | | Bridges |
| | | Main Roads | Secondary Roads | Impacted PPA ** | |
|---|---|---|---|---|---|
| 01 | Planned | 7.32 | 27.41 | 0.11 | 1 |
| | Implemented | 8.14 | 25.41 | 0.10 | 1 |
| | Tomlin | 12.09 | 26.62 | 0 | 0 |
| | Minimum Spanning tree | 14.14 | 32.20 | 0 | 0 |
| 02 | Planned | 7.99 | 30.11 | 0 | 0 |
| | Implemented | 8.54 | 26.50 | 0.15 | 1 |
| | Tomlin | 8.77 | 30.24 | 0 | 0 |
| | Minimum Spanning tree | 13.45 | 39.89 | 0 | 0 |
| 03 | Planned | 6.22 | 29.60 | 0 | 0 |
| | Implemented | 6.77 | 29.52 | 0 | 0 |
| | Tomlin | 8.99 | 33.23 | 0 | 0 |
| | Minimum Spanning tree | 15.04 | 39.86 | 0 | 0 |

* Annual Production Unit (APU) in the Caxiuanã National Forest. ** Permanent Protected Area (PPA).

The CEMAL timber company, which was authorized to conduct selective logging in the study site, effectively implemented secondary roads. These roads impacted a smaller area (25.41 hectares) of native forest than the area originally planned (27.41 hectares) by the timber company. Using the Tomlin and Minimum Spanning Tree modeling approaches, we projected that the direct impact on native forests would be 26.62 hectares and 32.2 hectares, respectively. This indicates that the Tomlin modeling approach yielded better results for planning secondary forest roads in the study site. Overall, the secondary road network field-implemented by the timber company achieved the best results in the Caxiuanã National Forests' management project. However, it impacted on some Permanent Protected Areas (PPA), where selective logging activities should not be carried out (Table 3).

Figure 6 provides a visual representation of the main and secondary roads as well as the log-landings planned and implemented by the CEMAL timber company in the Caxiuanã National Forest. Furthermore, two models, the Tomlin and Minimum Spanning Tree approaches, were applied and evaluated in this analysis. Both models encompassed the main road and secondary roads within the three Annual Production Units (APU) in the study site, as presented in Figure 6.

Similarly, we assessed the forest impacts (effective and predicted) of all modeling approaches. The originally planned main roads would impact 7.99 ha, while the main roads field-implemented by CEMAL Ltd. impacted a total of 8.54 ha. The Tomlin and the Minimum Spanning Tree modeling approaches predicted that total areas of 8.8 ha and 13.45 ha of forests would be impacted by the main road construction, respectively (Table 3, Figure 7). By comparing the four modeling approaches, the original planned road network showed better results. It is important to note that the Tomlin model showed similar results to the road network field-implemented by the timber company and did not impact any Permanent Protected Areas.

The originally planned secondary roads would impact a total of 30.11 ha of forest, while the field-implemented ones impacted a total of 26.50 ha. Meanwhile, the secondary road network predicted by the Tomlin model would impact a total of 30.24 ha, and the Minimum Spanning Tree model a total of 39.89 ha (Figure 7). By comparing all modeling approaches, the field-implemented road network showed the best results in the APU 02, although it impacted some Permanent Protected Areas (Table 3).

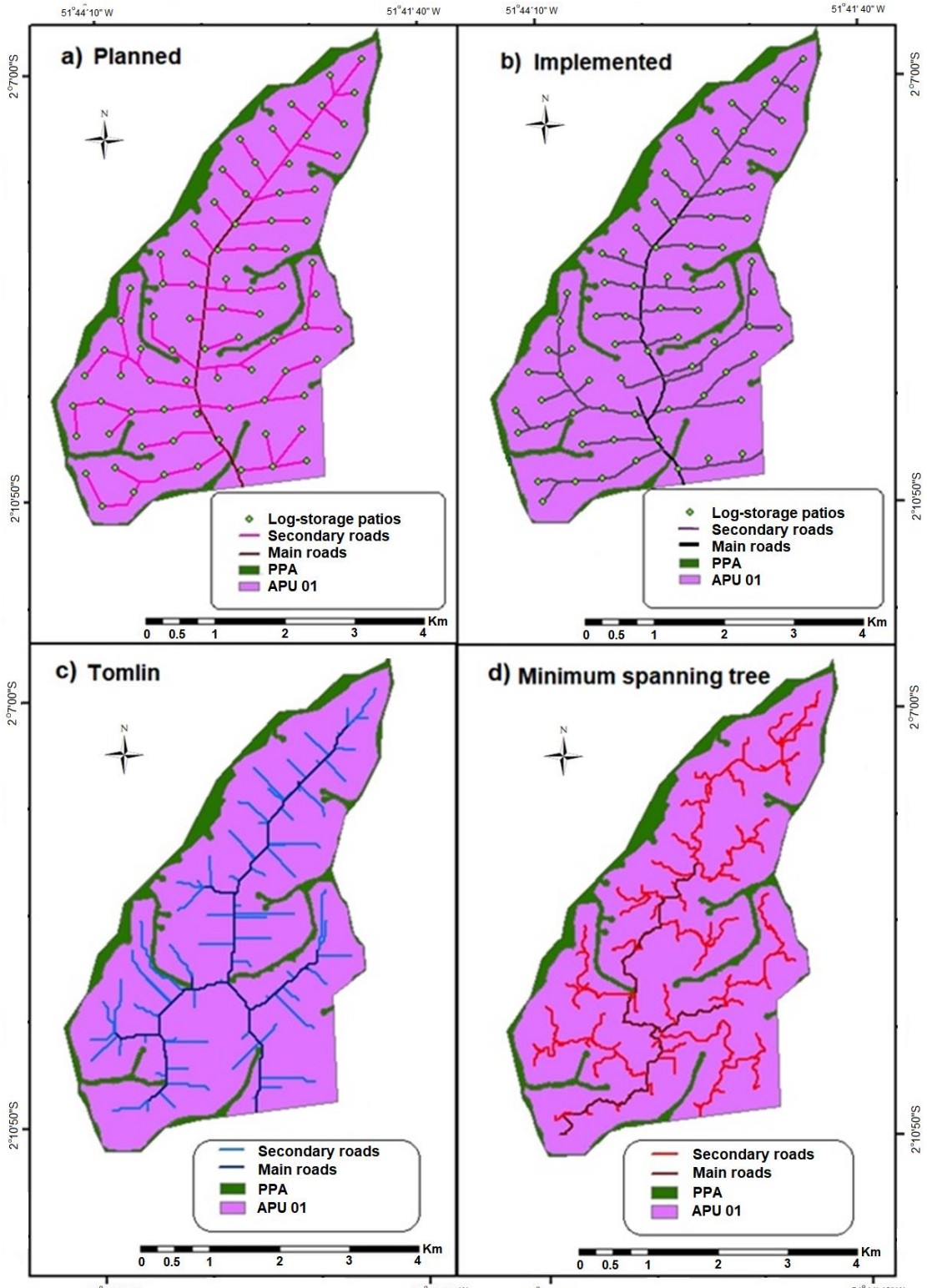

**Figure 6.** Annual Production Unit (APU) 01: (**a**) planned roads (main and secondary roads) and log-landings; (**b**) implemented roads (main roads, secondary roads) and log-landings; (**c**) modelled roads using the TOMLIN modeling approach with Tobler´s hiking cost, including main and secondary roads; (**d**) modelled roads using the Minimum Spanning Tree with Dheight friction cost, including mains and secondary roads. PPA = Permanent Protected Areas in which selective logging activities are not allowed and APU = Annual Production Unit, an area.

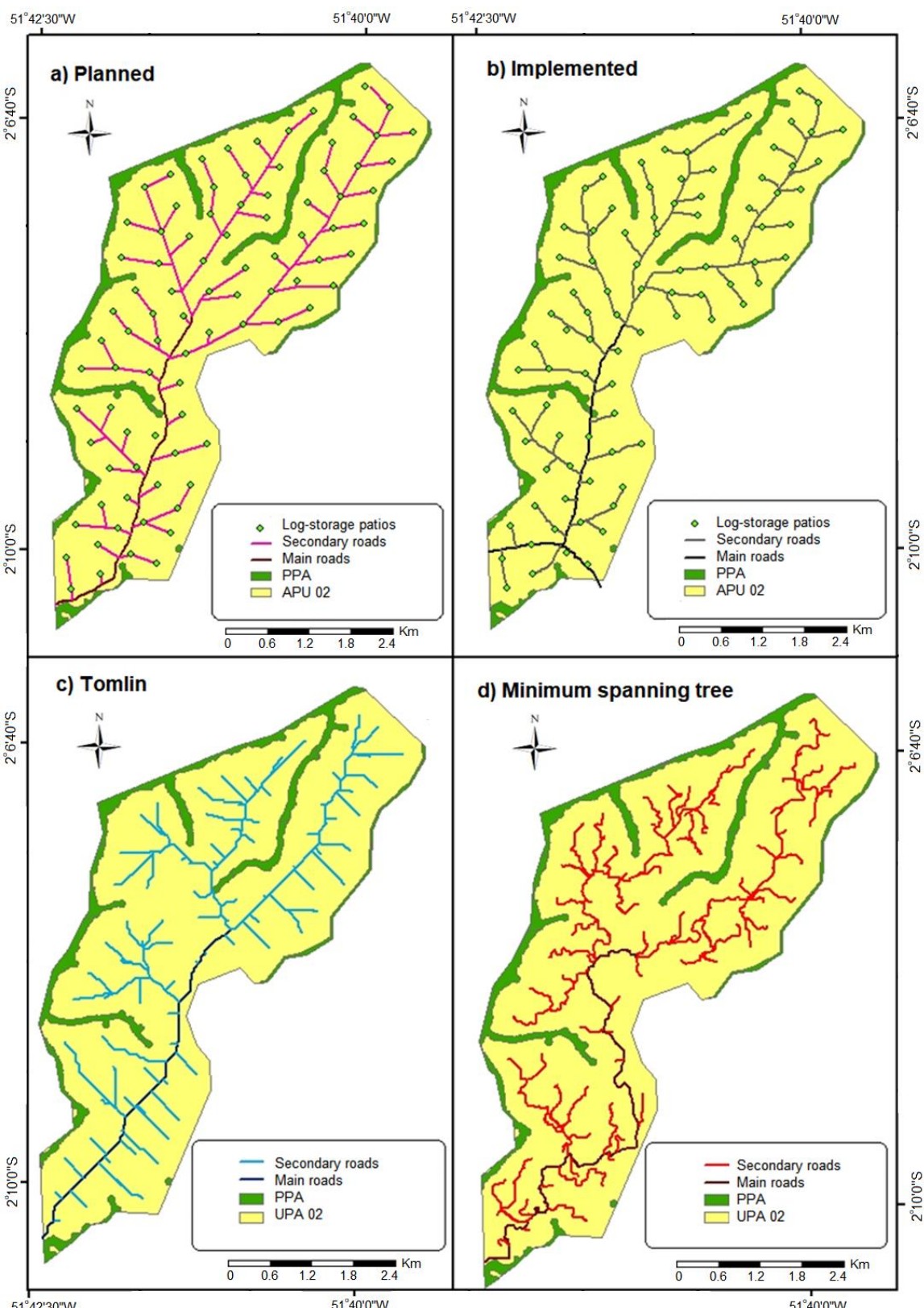

**Figure 7.** Annual Production Unit (APU) 02: (**a**) planned roads (main road, country road, and planned log-storage patios); (**b**) roads executed (main road, secondary road, and executed). (**c**) Roads delineated by Model 1 (TOMLIN) contain main roads and secondary roads. (**d**) Roads modelled by Model 2 (Minimum Spanning Tree) contain main roads and secondary roads.

In the APU 03, the originally planned main roads would impact 6.22 ha, while the main roads field-implemented by CEMAL Ltd. impacted a total of 6.77 ha. The Tomlin and the Minimum Spanning Tree modeling approaches predicted that a total area of 8.99 ha and 15.04 ha of forests would be impacted by the main road construction, respectively (Table 3, Figure 8). By comparing the four modeling approaches, the field-implemented main road network showed slightly better results.

The originally planned secondary roads would impact a total of 29.60 ha of forests while the field-implemented ones impacted a total of 29.52 ha. Meanwhile, the secondary road network predicted by the Tomlin model would impact a total of 33.23 ha, and that by the Minimum Spanning Tree model a total of 39.86 ha. By comparing all modeling approaches, the field-implemented road network showed the best results in the APU 03.

Table 3 shows the number of bridges and the PPA affected by each modeling approach. A bridge was planned and constructed in APU 01, and a bridge was constructed in APU 02, effectively impacting a total of 0.1 and 0.15 ha of forests, respectively. By applying the Tomlin and Minimum Spanning Tree approaches, no bridges would be required in the study area, and consequently no APP would be affected.

In the last iteration, the Tomlin modeling created parallel segments which were not considered optimum for road design and, consequently, for field-implementation. The model was not developed to perform the last iteration, so it is considered ideal only for modeling main roads only, showing low performance for secondary road modeling.

When this algorithm (Tomlin) is implemented, it records all iterations. We observed that in the penultimate iteration it had not generated the parallel segments, so not all log-landings would be connected in the road network (Figure 9). By estimating the least cost path with more efficient roads, it is recommended to use a combination of the two proposed models. Using the last iteration of Tomlin as the source in the application of the Minimum Spanning Tree model helps to prevent the design of parallel roads.

In the Minimum Spanning Tree modeling, utilizing the Dheight friction cost would result in outcomes more closely aligned with those implemented by the concessionaire company. The implemented model demonstrated lower economic costs since it did not avoid crossing PPAs (Permanent Protected Areas), unlike the modeled design, which bypassed such areas when encountered.

Finally, we observed that the Tomlin model could support the definition of forest operation plans, which must be annually reviewed by forest concessionaires. According to [17], the annual report must be submitted every year by a forestry professional containing all forest operation planning for the area of interest. Before any forest activity is officially authorized, the annual forest operation plan must be analyzed and approved by an environmental agency in Brazil. Both the forest management plan and annual forest operation plan are considered legal requirements for selective logging in native forests in Brazil (Normative Instruction no. 05/2006 Ministry of the Environment—MMA, and Resolution no. 406/2009 National Council of Environment—CONAMA).

### 3.2. Road Modeling Based on the Location of the Selectively Logged Trees

In addition, we employed the Tomlin and Minimum Spanning Tree modeling approaches for forest road modeling at the tree location level. In this phase, we utilized the geographic coordinates of the trees that were effectively logged by the timber company as the starting points, in conjunction with the previously determined geographic destinations in this analysis. The infrastructure field-implemented by the CEMAL timber company, including main roads, secondary roads, and skid trails, as well as the theoretical infrastructure derived from the Tomlin and Minimum Spanning Tree modeling approaches, are depicted in Figures 10–12.

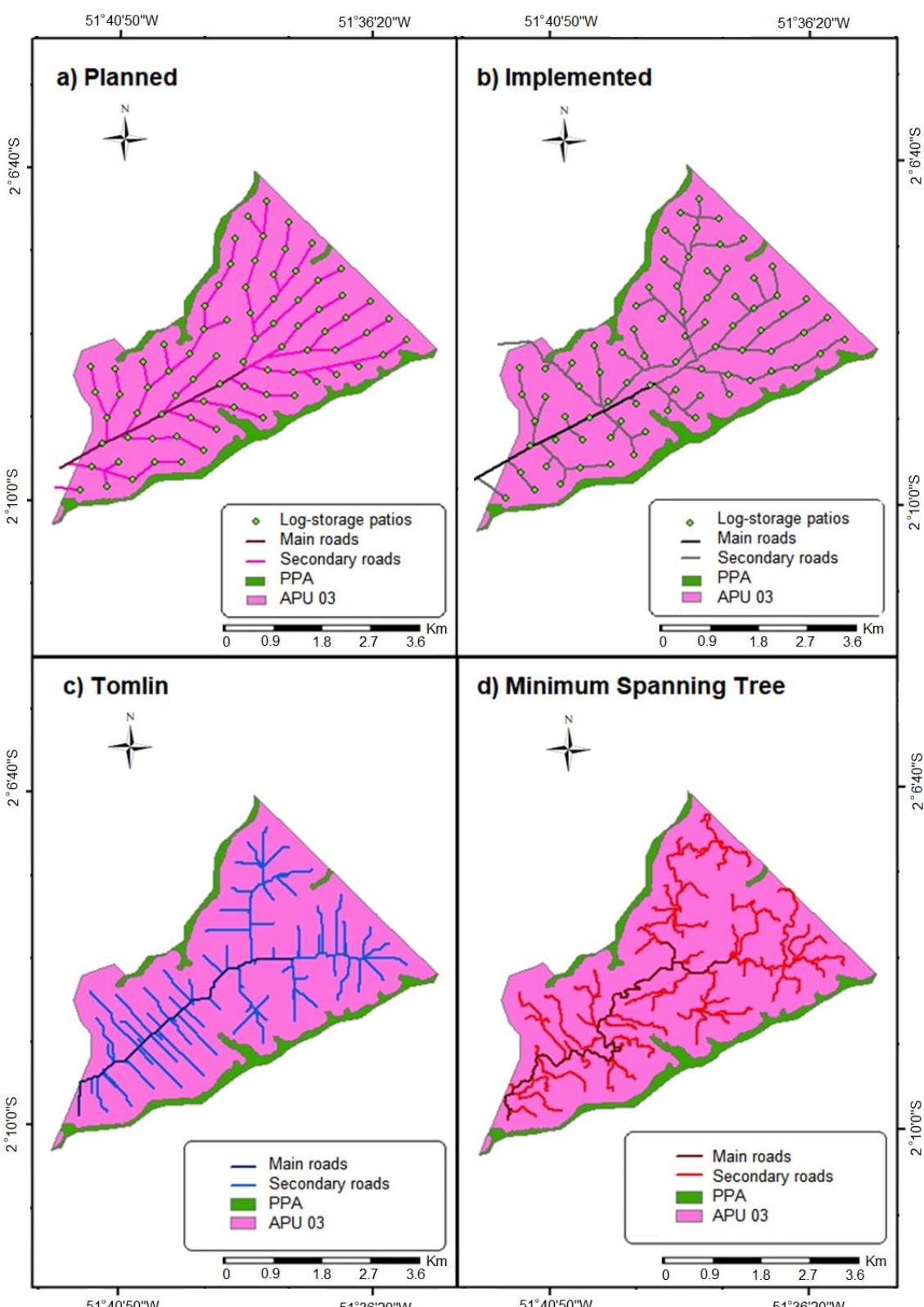

**Figure 8.** Annual Production Unit (APU) 03: (**a**) planned roads (main road, secondary road, and planned patios); (**b**) roads executed (main road, secondary road, and executed). (**c**) Roads delineated by Model 1 (TOMLIN) contain main road and secondary roads. (**d**) Roads modelled by Model 2 (Minimum Spanning Tree) contain main roads and secondary roads.

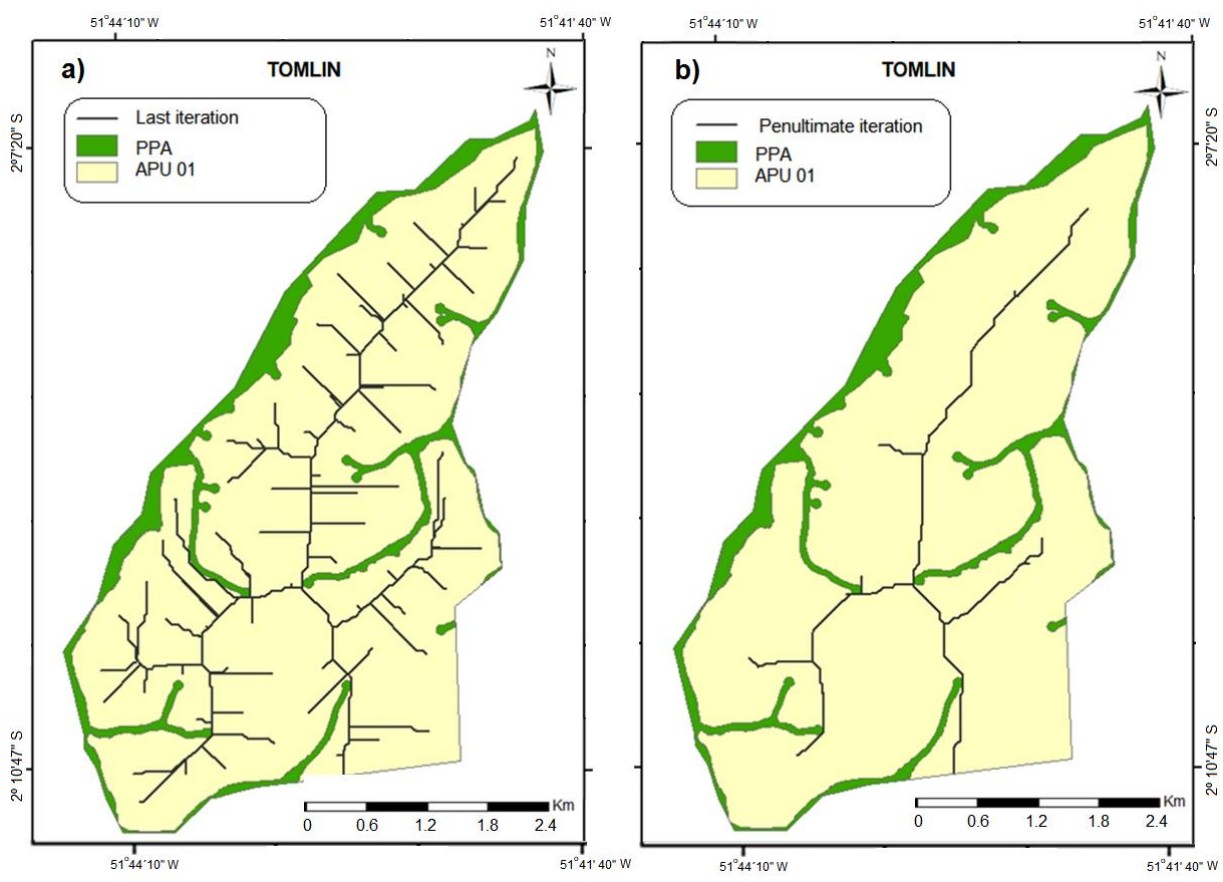

**Figure 9.** (**a**) Tomlin modeling with last iteration and (**b**) the Tomlin modeling with the penultimate iteration.

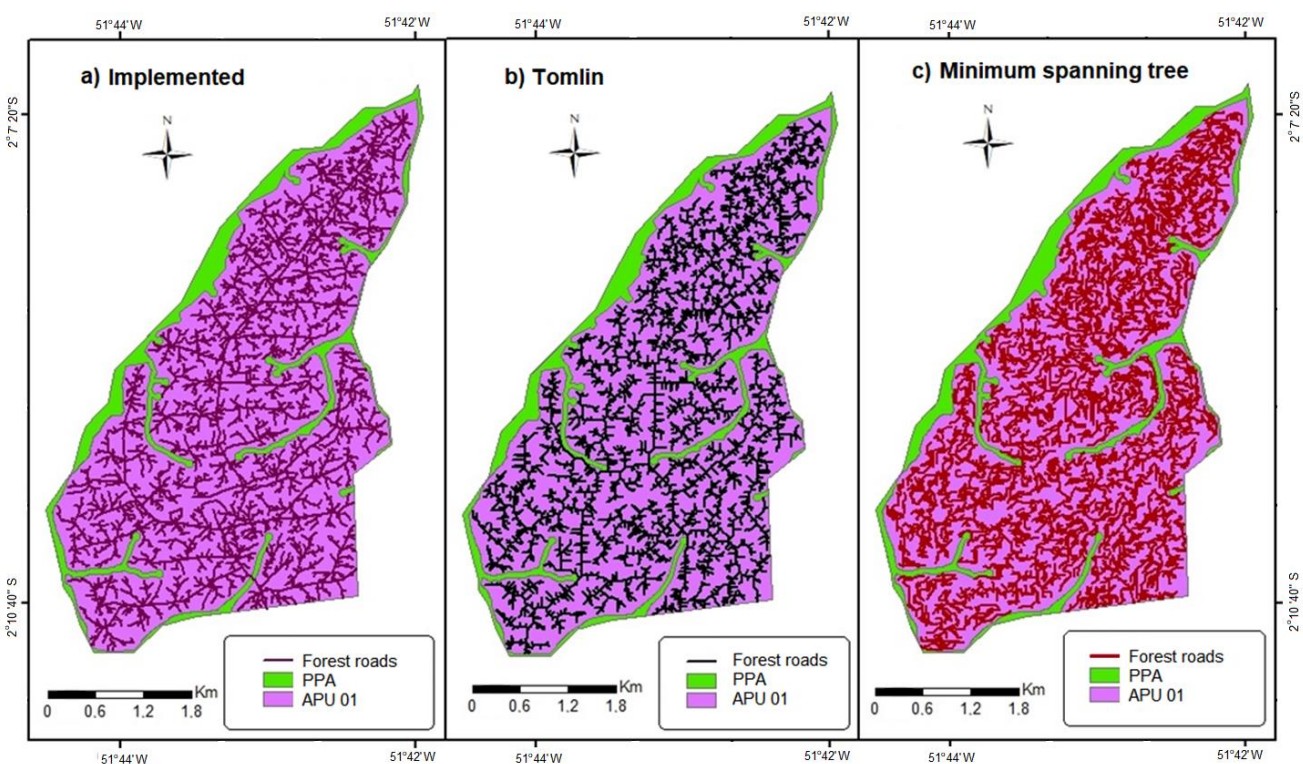

**Figure 10.** APU 01: (**a**) logging infrastructure implemented by CEMAL Ltd.; (**b**) road design by applying Model 1 (TOMLIN) and (**c**) Model2 (Minimum Spanning Tree).

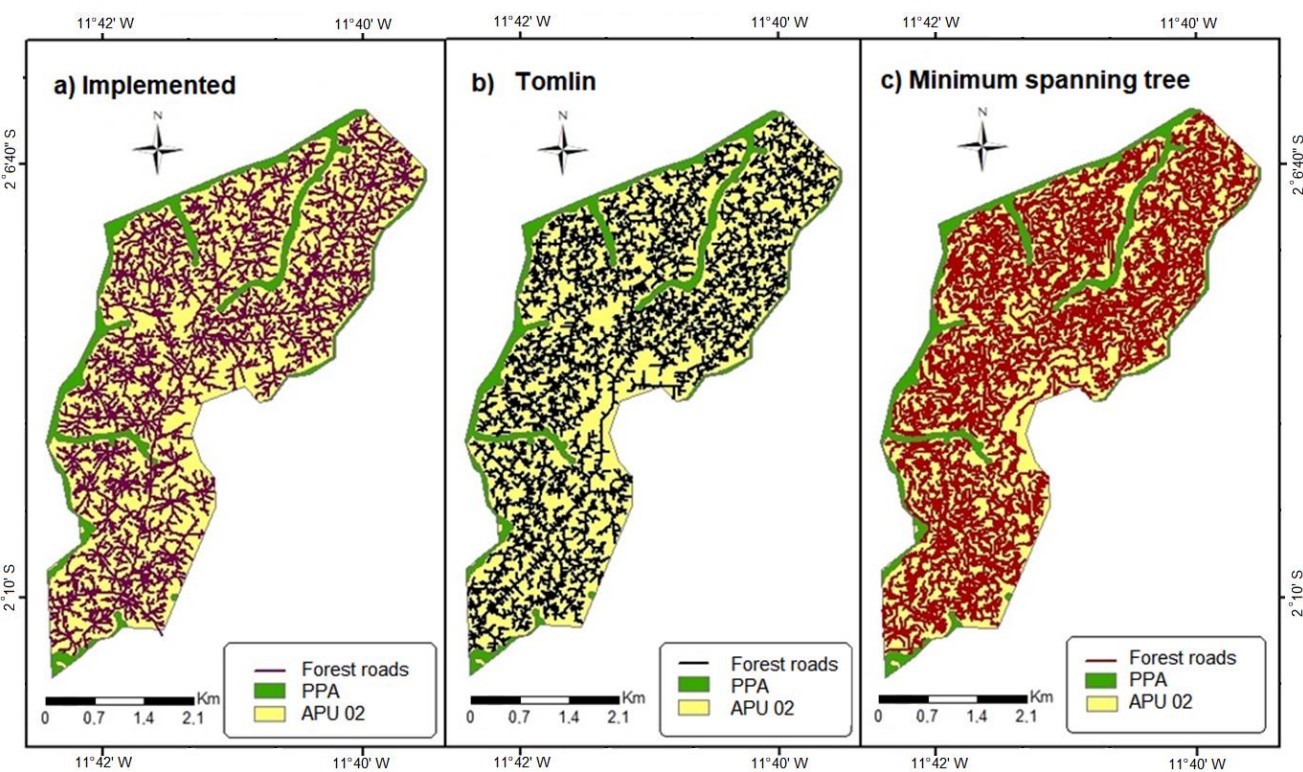

**Figure 11.** APU 02: (**a**) logging infrastructure implemented by CEMAL Ltd.; (**b**) road design by applying Model 1 (TOMLIN) and (**c**) Model2 (Minimum Spanning Tree).

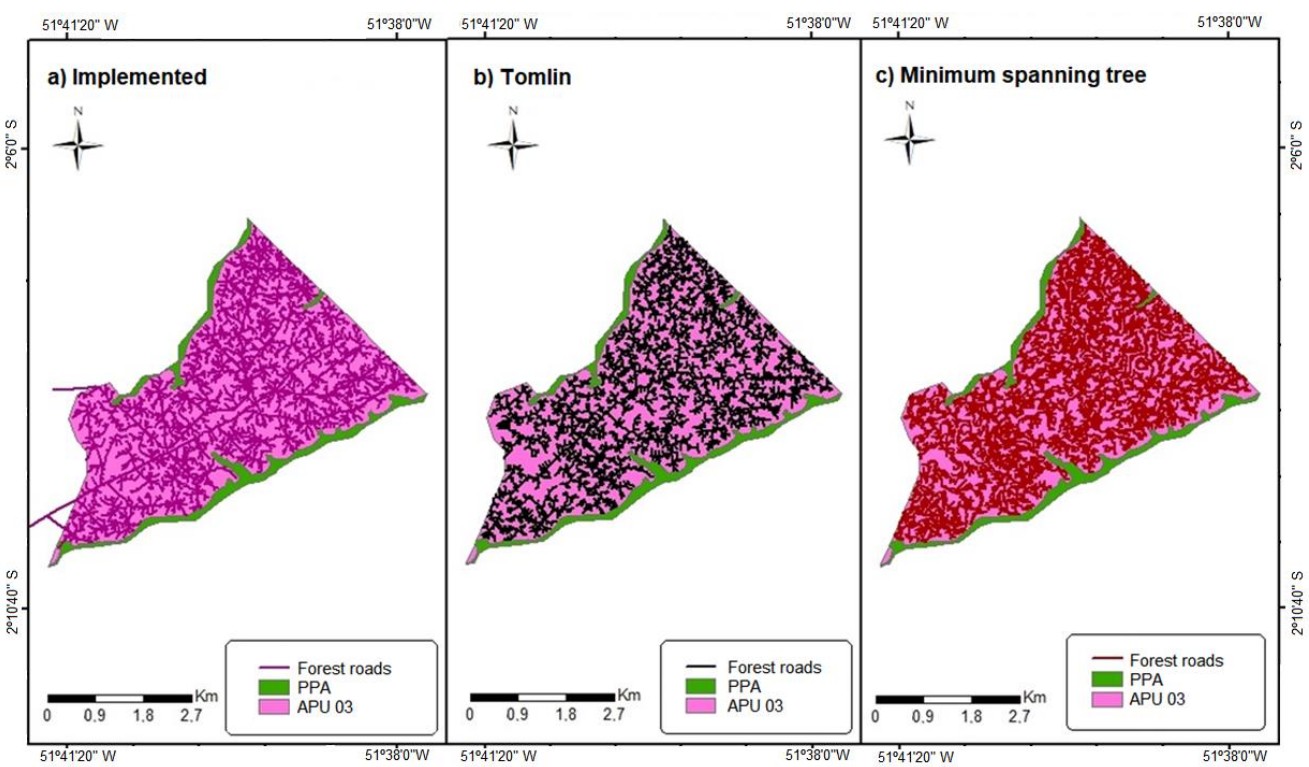

**Figure 12.** APU 03: (**a**) logging infrastructure implemented by CEMAL Ltd.; (**b**) road design by applying Model 1 (TOMLIN) and (**c**) Model2 (Minimum Spanning Tree).

The applied Tomlin model showed results similar to the infrastructure field-implemented by the concessionaire timber company when comparing the perimeters (length in meters) of the modelled and constructed roads and skid-trails (Table 4).

**Table 4.** The total length of roads and skid trails.

| APU * | Total Length of Roads and Skid Trails (Meters) | | |
|---|---|---|---|
| | **Field-Implemented** | **Tomlin** | **Minimum Spanning Tree** |
| 01 | 232,640.1 | 249,376.7 | 431,715.2 |
| 02 | 286,268.0 | 324,140.0 | 564,977.3 |
| 03 | 266,895.6 | 283,017.1 | 498,923.0 |

\* Annual Production Unit.

### 3.3. Optimum Road Density (ORD) Determination

The current, optimum, and acceptable costs of Road Density (RD) are described in Tables 5–7. When comparing the all-road densities in APU 01, we observed that the implemented infrastructure demonstrated the best performance, achieving a result of 17.48 m·ha$^{-1}$. This value represents the smallest road density among the assessed models. the Tomlin model obtained a result of 18.47 m·ha$^{-1}$, which was close to the value calculated acceptable value of 18.41 m·ha$^{-1}$. The lower road density corresponds to fewer areas of infrastructure resulting in reduced environmental impacts on the forest.

**Table 5.** APU 01: costs of selective logging, roads, environmental degradation, total cost, road density, average road density, and spacing between secondary roads with the current values, in addition to the optimum and acceptable values for each variable.

| Modeling Approach | Cext | Croad | Cdeg | CTot | RD | ASD | RS |
|---|---|---|---|---|---|---|---|
| | USD m$^{-3}$ | | | | m·ha$^{-1}$ | m | m |
| Planned | 8.33 | 4.48 | 1.40 | 14.21 | 18.83 | 248.94 | 573.15 |
| Field-implemented | 8.52 | 4.16 | 1.30 | 13.98 | 17.48 | 268.16 | 607.03 |
| Tomlin model | 8.38 | 4.40 | 1.37 | 14.15 | 18.47 | 253.79 | 589.41 |
| Spanning tree model | 7.93 | 5.36 | 1.67 | 14.97 | 22.51 | 208.24 | 589.41 |
| Optimum | 9.68 | 2.89 | 0.90 | 13.47 | 12.11 | 386.98 | 825.37 |
| Acceptable | 8.39 | 4.38 | 1.37 | 14.14 | 18.41 | 254.62 | 543.18 |

Cext: forest extraction cost; Croad: cost of roads; Cdeg: cost of degradation; CTot: total cost; RD: road density; ASD: average skid distance; and RS: spacing between roads.

**Table 6.** APU 02: costs of selective logging, roads, environmental degradation, total cost, road density, average road density, and spacing between secondary roads with the current values, in addition to the optimum and acceptable values for each variable.

| Modeling Approach | Cext | Croad | Cdeg | CTot | RD | ASD | RS |
|---|---|---|---|---|---|---|---|
| | USD m$^{-3}$ | | | | m·ha$^{-1}$ | m | m |
| Planned | 8.27 | 4.61 | 1.44 | 14.32 | 19.36 | 242.12 | 569.21 |
| Field-implemented | 8.59 | 4.06 | 1.27 | 13.91 | 17.05 | 274.93 | 589.41 |
| Tomlin model | 8.25 | 4.65 | 1.45 | 14.34 | 19.51 | 240.26 | 597.75 |
| Spanning tree model | 7.66 | 6.20 | 1.93 | 15.79 | 26.03 | 180.08 | 597.75 |
| Optimum | 9.68 | 2.89 | 0.90 | 13.47 | 12.11 | 386.98 | 825.37 |
| Acceptable | 8.39 | 4.38 | 1.37 | 14.14 | 18.41 | 254.62 | 543.18 |

Cext: forest extraction cost; Croad: cost of roads; Cdeg: cost of degradation; CTot: total cost; RD: road density; ASD: average skid distance; RS: spacing between roads.

Acceptable Road Spacing (ARS) and Optimal Road Density (ORD) are related to Road Density (RD). A road being more spaced from another means that the road network is smaller, which increases the cost of extraction; however, it lowers the cost of environmental degradation and the cost of building and maintaining roads. Ideally, models that come close

to ORD and ARS values are the most appropriate spacings. Based on this, the spacings of all models assessed in this analysis for the APU 01 were considered optimum or acceptable.

**Table 7.** APU 03: costs of selective logging, roads, environmental degradation, total cost, road density, average road density, and spacing between secondary roads with the current values, in addition to the optimum and acceptable values for each variable.

| Modeling Approach | Cext | Croad | Cdeg | CTot | RD | ASD | RS |
|---|---|---|---|---|---|---|---|
| | USD m$^{-3}$ | | | | m·ha$^{-1}$ | m | m |
| Planned | 8.30 | 4.54 | 1.42 | 14.26 | 19.06 | 245.93 | 732.39 |
| Field-implemented | 8.31 | 4.52 | 1.41 | 14.25 | 19.00 | 246.71 | 1190.97 |
| Tomlin model | 8.00 | 5.18 | 1.62 | 14.81 | 21.77 | 215.32 | 960.47 |
| Spanning tree model | 7.66 | 6.18 | 1.93 | 15.78 | 25.97 | 180.5 | 960.49 |
| Optimum | 9.68 | 2.89 | 0.90 | 13.47 | 12.11 | 386.98 | 825.37 |
| Acceptable | 8.39 | 4.38 | 1.37 | 14.14 | 18.41 | 254.62 | 543.18 |

Cext: forest extraction cost; Croad: cost of roads; Cdeg: cost of degradation; CTot: total cost; RD: road density; ASD: average skid distance; and RS: spacing between roads.

In the APU 02, the field-implemented road density (RD) showed the best results, a total of 17.05 m·ha$^{-1}$, which was a length close to the acceptable value of 18.41 m·ha$^{-1}$. By comparing the applied models in this analysis, the best RD results were observed for the Tomlin modeling, which estimated a total of 19.51 m·ha$^{-1}$ compared to 26.03 m·ha$^{-1}$ derived from the Minimum Spanning Tree modeling approach. Nevertheless, the road spacing of all assessed models in this study were considered optimal or acceptable for the APU 02.

We observed that the planned and field-implemented road densities (RD) were slightly closer to the acceptable RD for the APU 03. The Tomlin model showed better results compared to the Minimum Spanning Tree model and was also close to the acceptable RD for APU 03. The field-implemented road network demonstrated similar RD results to the Tomlin model, measuring 19.0 m·ha$^{-1}$.

Figure 13 presents a visual depiction of the changes in road construction and maintenance costs, skidder extraction costs, environmental degradation costs, and the overall cost as road density increases. The figure illustrates the optimal and acceptable road densities in this analysis, based on the assumption of achieving the minimum total cost at the inflection point of the Cglobal curve. This point represents the optimum road density (12.1 m/ha) identified in the study site.

This analysis (Figure 13) indicates that adopting better planning for forest infrastructure will result in a reduction in financial costs associated with selective logging. While the reduction per harvesting unit may be relatively low, it has the potential to significantly improve the overall cost reduction.

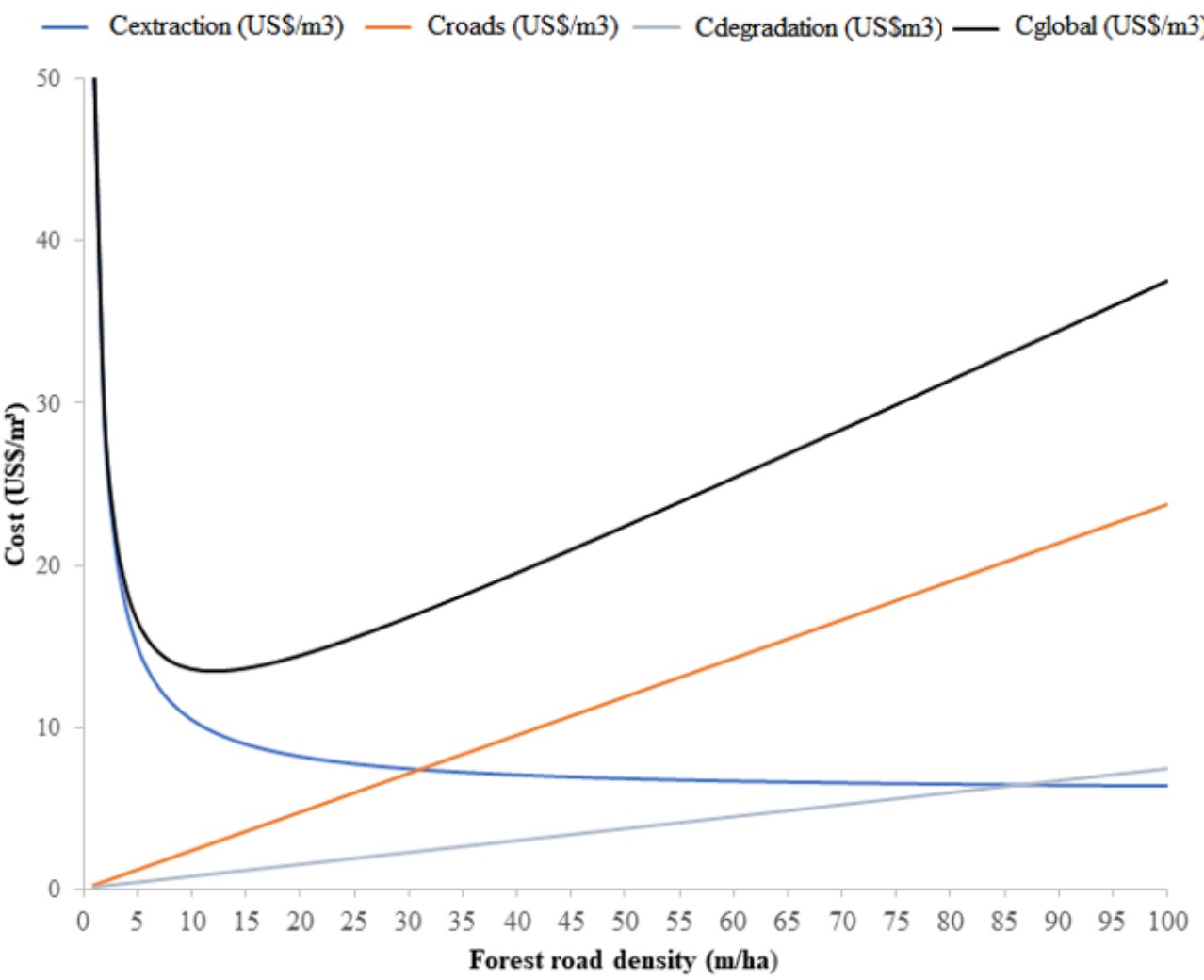

Cextraction: Extraction cost; Croads: Cost of roads; Cdegradation: Cost of degradation; Cglobal: Global cost.

**Figure 13.** Relationship between road density and road costs, extraction and degradation costs, and loss of productive forest area in a clear-cutting system.

### 4. Discussion

Based on Tables 5–7, we observed a strong relationship between road density (RD) and road spacing (RS), which plays a crucial role in field planning. Additionally, road density is also connected to the average skid distance (ASD), indicating that these factors significantly impact the costs of selective logging operations. Even a slight cost reduction, such as USD 1 per meter, should be properly considered by timber companies. For instance, the optimal road density of 12.1 m/ha corresponds to an RS of 825.37 m and an ASD of 386.98 m (Table 7).

When analyzing the forest road network in the study area, the forest density is an important factor that must be properly considered because it is related to the extraction method, road density, and potential permanent damage to the forest landscape. Therefore, the selective logging method will determine a higher or lower road density, and the characteristics and particularities of each piece of equipment and its optimized use will point out the best economic alternatives for each area of interest [18].

The determination of the optimal density of roads (ORD) is a quantitative method to establish the volume of roads in each forest region, optimizing the road/forest har-

vest ratio in technical and economic terms [35]. In their study, [35] estimated a total of 26 m·ha$^{-1}$ as the optimal road density, which is higher than the value estimated in this study (12.1 m·ha$^{-1}$). These authors calculated the spacing between roads as 385 m, which is substantially lower than the value found in this analysis (825.37 m). Estimating the optimal road density is often a challenging task because local conditions in an area of interest and operational constraints may not allow effective construction and implementation [37].

Theoretically, the distance between two log-landings should not be less than the Optimum Distance among roads. This definition can be used as a reference for the allocation of patios since the log grouping can deface certain points or add others. However, it helps to optimize the log skid and, consequently, reduce logging costs [19].

The final road density of a road network is correlated with the means of production used in the logging operations; subjectively, the higher the road density, the lower the average skid distance. Finally, what is more relevant is the sum of road construction costs and extraction costs, always seeking to opt for the minimum total cost and lesser environmental disturbances [34].

In Figure 13, the intersections between the blue line (extraction cost) and the orange line (cost of roads), as well as the intersection between the blue line (extraction cost) with the gray line (cost of environmental degradation), are the maximum densities tolerated; after these points, the economic and environmental impact is greater than the amount of roads required. The line crossing can be interpreted as the technical definition, and the overall cost as the economic definition for road planning. The overall cost analysis also indicates that the optimal road density is 12.1 m/ha (Figure 13), considering the cost of extraction and environmental impacts (cost of degradation) by comparing the modifications in logging infrastructure and their effects on the forest.

## 5. Conclusions

Road construction using theoretical models, as analyzed in this study, showed higher lengths than the roads field-implemented by the concessionaire timber company in the Caxiuanã National Forest, state of Pará, Brazil. However, the road networks designed by the models showed a potential to cause fewer forest impacts and disturbances by not affecting any Permanent Protection Areas in the study site, keeping the construction cost at values close to the field-implemented ones and optimizing the track location in the field.

The Tomlin model showed the best economic and environmental performance in modeling road density and allocation in the study area. By analyzing the models based on the trees as the destination, we observed that neither the Minimum Spanning Tree nor the Tomlin modeling approaches performed well when considering the length of the skid trails, with solutions apparently economically unfeasible considering the pattern of high-density roads compared with those effectively constructed in the study area. Processing capability was also an important constraint when using trees as the destination. As for road density costs, the infrastructure field-implemented by the timber company showed the best results for the three APUs in the study site.

We strongly recommend using the Tomlin modeling approach for the main forest road modeling and, subsequently, applying the penultimate iteration of Tomlin as the source and running the data again using the Minimum Spanning Tree to delineate the secondary roads. For best results, we recommend using the log-landings instead of commercial trees as the final destinations in this modeling approach.

By developing road planning using the modeling that showed better results in this analysis, we also introduced an advantage of geoprocessing to support forest management planning. The Tomlin model proved to be feasible and useful to plan the Annual Operational Plan in forest management in tropical forests. Finally, based on our results, we conclude that Minimum Spanning Tree modeling can be successfully applied to outline skid trails in the study site.

Opportunities for future studies include utilizing larger forest datasets to assess the geoprocessing capabilities of the Python script using ArcPro's libraries. Additionally, it

would be interesting to test the proposed modeling approaches in more heterogeneous study areas.

**Author Contributions:** Conceptualization, P.d.P.M., E.Y.A. and Á.N.d.S.; methodology and validation, P.d.P.M., E.Y.A., Á.N.d.S. and R.S.P.; formal analysis, P.d.P.M., E.Y.A., R.M.C., F.E., Á.N.d.S., R.S.P. and E.A.T.M.; investigation and data curation, P.d.P.M., R.M.C., E.Y.A., F.E., Á.N.d.S., R.S.P., E.A.T.M. and E.P.M.; writing—preparation of original draft, P.d.P.M.; writing—review and editing, P.d.P.M., Á.N.d.S., E.Y.A. and E.A.T.M. All authors have read and agreed to the published version of the manuscript.

**Funding:** This research was funded by the National Council for Scientific and Technological Development (CNPq), Grants no. 401892/2021-2 and no. 311155/2020-0). Coordenação de Aperfeiçoamento de Pessoal de Nível Superior; Award no. Finance Code 001.

**Institutional Review Board Statement:** Not applicable.

**Informed Consent Statement:** Not applicable.

**Data Availability Statement:** The dataset will be available upon direct request to the corresponding author.

**Acknowledgments:** We thank the CEMAL timber company for providing access to the logging site and field data for this research.

**Conflicts of Interest:** The authors declare no conflict of interest. The funders had no role in the design of the study; in the collection, analyses, or interpretation of data; in the writing of the manuscript; or in the decision to publish the results.

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
