# Peer review of "Assessment of Forest Road Models in Concession Areas in the Brazilian Amazon"

_forests, doi:10.3390/f14071388_

Round 1

Reviewer 1 Report

Dear authors

It is an interesting research. Please revise the following items in the article.

Abstract

Lines 14-23: I think it is long. One sentence is enough for the importance of the research topic.

Lines 28-31: Please write the results in more detail: economic comparison and environmental comparison

Introduction

The method of referring to the references inside the text, as well as the references at the end of the article, should be adjusted according to the Journal format.

Lines 38-101: This part consists of eight paragraphs, which seems long. It is better to summarize the local, regional, and international importance of the studied forests in one paragraph. In another paragraph, the method of managing these forests (silviculture and logging systems) in relation to forest roads should be stated.

By reading the introduction, the problem or research question was not identified. It is better to point out the shortcomings and possible defects of the previous forest road design methods and the superior features of the new forest road design methods.

In my opinion, it will be clearer if refer to the total area of the country's forests, the area of forests with a management plan, the area of forests that require the design and construction of forest roads in order to extract wood.

Line 82: Please more explain about “concession forest”, “forest concession” “concession forest. 

Line 104: In my opinion, the word "log landings" is more familiar to the readers than the word "log-storage patios".

Line 106-108: This sentence is not clear to me. What do you mean by the word "impact"? Please explain more. Is your mean “soil disturbance” or “residual tree/regeneration damage”?

Line 112: "log landings" instead of (log-storage patios).

Line 128: “The” instead of “the”.

Materials and methods

Study site

Line 137: What is your mean by “Flona”?

Line 143: What is your mean by “Annual Production Unit”?

Table 1:

Are there any other differences between the three studied areas except for the area?

Why the geographic, physiographic, forestry and exploitation characteristics of these areas are not included in this table?

It seems that zero is not necessary before the number: "1" should be used instead of "01".

Instead of the word hectares, its abbreviation "ha" should be used.

Figure 1:

What is differences between APU 1, APU 2, and APU 3?

Line 157: What is your mean by “Am”?

Line 158: It doesn't seem necessary to rewrite "temperature".

Line 166: It seems that a more complete description of the studied forests should be provided: Tree species, tree height and diameter, stand structure.

Figure 2: The position of harvested trees in the forest is not clear. It will look better if the difference between harvested trees and remaining trees is shown with different colors.

Line 211 and Figure 3: After felling instead of “after logging”.

Line 313: (1) instead of (01)

314: RD road density

Line 319: 1.85 for V or for V*T?

Line 369 and 370: USD/L and USD/h

Line 423: Eq 11: what is constant?

Line 428: please correct: “equation”

Figure 6: Please remove the extra b)” in figure 6-b.

Figure 6C: The colors of the main road and secondary road in the figure have been replaced by their color in the figure legend. It seems to need correction.

Line 578: What is your mean by “Dheight cost”? Please make it more clear.

Line 585-589 and line 600-602: The results of this review are not clear.

Table 4: Title: “length of the skid trails” or “total length of roads and skid trails”?

Tables 5 and 6: “Planned” and values are bold and underlined, please correct it.

Figure 13: what is results from this figure? Is it for all three study areas?

Line690: How is this value calculated and in which figure or table is it mentioned?

Line 690: Please make the discussion more complete, especially environmental discussions.

Author Response

Thank you for your careful, insightful, and useful comments. We have taken all points into serious consideration and have incorporated changes to the manuscript and included a response below. We are grateful to the editor and reviewers for substantially improving this manuscript. Here addressed the R1 comments in general and specifically.

Please find our responses in gray font color in our revision document to make it easier for the Editor and reviewer to see the edits. 

Please, refer to the attached *.doc file.

Reviewer 2 Report

I have found the topic of your research very interesting. The analysis of your methodology and results was very satisfying, and I found it particularly interesting that you developed elements of spatial analysis into an immediate actionable object such as the forest road planning.

I have only some minor comments, which I believe they would improve your work even more.

If it is possible, it would be useful if you could add some information in the 2.1. Study site section, regarding the main forest species and the time periods of logging.  

One of the key parameters for the application was the digital elevation model. Could you give a general description of the topography of the area? e.g. if there is strong or mild relief.

In section 2.3. Data preprocessing, you write in lines 292 and 293, that the algorithms were implemented using ArcPro’s libraries. If it possible please clarify this statement, and be more accurate regarding the GIS tools you used in order to create these algorithms.

In the Conclusions section, please add some information about problems you encountered. The primary data you collected was of sufficient quality? In addition, it would be good to propose some future research on the subject.

Finally, in the introdution section you do not present any extent literature review of similar topics. It would useful to add some literature review of similar recent researches, regarding the spatial analysis of forest roads and comment about their aproach relative to yours. I suggest you to look and add in your references if you find it interesting the:

Kolkos, G., Stergiadou, A., Kantartzis, A. et al. Effects of forest roads and an assessment of their disturbance of the natural enviroment based on GIS spatial multi-criteria analysis: case study of the University Forest of Taxiarchis, Chalkidiki, Greece. Euro-Mediterr J Environ Integr (2023). https://doi.org/10.1007/s41207-023-00362-6

I have found your English good. I only reccomend proof reading in order to avoid any mistakes.

Author Response

Thank you for your careful, insightful, and useful comments. We have taken all points into serious consideration and have incorporated changes to the manuscript and included a response below. We are grateful to the editor and reviewers for substantially improving this manuscript. Here addressed the R2 comments in general and specifically.

Please find our responses in gray font color in our revision document to make it easier for the Editor and reviewer to see the edits.

Please, refer to the attached *.doc file that contains our point by point responses.
